EMBO
reports

# scientific report

# Adenosine-A$_3$ receptors in neutrophil microdomains promote the formation of bacteria-tethering cytonemes

Ross Corriden[1], Tim Self [1], Kathryn Akong-Moore[2], Victor Nizet[2,3], Barrie Kellam[4], Stephen J. Briddon[1] & Stephen J. Hill[1+]

[1]Institute of Cell Signalling, School of Biomedical Sciences, Medical School, University of Nottingham, Nottingham, UK, [2]Department of Pediatrics, [3]Skaggs School of Pharmacy and Pharmaceutical Sciences, University of California, San Diego, La Jolla, California, USA, and [4]Centre for Biomolecular Sciences, School of Pharmacy, University of Nottingham, Nottingham, UK

**The A$_3$-adenosine receptor (A$_3$AR) has recently emerged as a key regulator of neutrophil behaviour. Using a fluorescent A$_3$AR ligand, we show that A$_3$ARs aggregate in highly polarized immunomodulatory microdomains on human neutrophil membranes. In addition to regulating chemotaxis, A$_3$ARs promote the formation of filipodia-like projections (cytonemes) that can extend up to 100 μm to tether and 'reel in' pathogens. Exposure to bacteria or an A$_3$AR agonist stimulates the formation of these projections and bacterial phagocytosis, whereas an A$_3$AR-selective antagonist inhibits cytoneme formation. Our results shed new light on the behaviour of neutrophils and identify the A$_3$AR as a potential target for modulating their function.**
Keywords: neutrophils; adenosine receptors; host–pathogen interactions

## INTRODUCTION

Extracellular nucleotides/nucleosides are critical mediators of immune cell function [1]. Extracellular ATP has been proposed to act as a long-range 'find me/eat me' signal that promotes phagocytic clearance by leukocytes [2]. Indeed, extracellular ATP serves as an important autocrine/paracrine mediator of chemotaxis in several cell types [3,4]. In particular, polarized release of ATP in response to chemoattractant stimulation has a critical role in neutrophil chemotaxis [3]. Although several studies have concluded that ATP itself does not act as a neutrophil chemoattractant [3,5], the removal of endogenously released ATP

[1]Institute of Cell Signalling, School of Biomedical Sciences, Medical School, University of Nottingham, Nottingham NG7 2UH, UK
[2]Department of Pediatrics,
[3]Skaggs School of Pharmacy and Pharmaceutical Sciences, University of California, San Diego, La Jolla, California 92093-0687, USA
[4]Centre for Biomolecular Sciences, School of Pharmacy, University of Nottingham, Nottingham NG7 2UH, UK
[+]Corresponding author. Tel: +44 (0)115 82 30082; Fax: +44 (0)115 82 30081;
E-mail: stephen.hill@nottingham.ac.uk

by the enzymatic scavenger apyrase nevertheless inhibits the migration of these cells in response to chemoattractants, such as fMet-Leu-Phe (fMLP) [3]. Although the numerous potential fates of extracellular ATP [6] have made it difficult to pinpoint its precise role in the control of chemotaxis, inhibition of fMLP-mediated chemotaxis by addition of exogenous adenosine deaminase suggests that G-protein-coupled adenosine receptors are involved in the process [3].

On release, ATP is rapidly converted to adenosine by ecto-nucleotidases on the neutrophil cell surface [7], leading to activation of G-protein-coupled adenosine receptors. One member of this receptor family, the A$_3$-adenosine receptor (A$_3$AR), stands out as a key regulator of neutrophil function. A$_3$AR-knockout mice show elevated bacterial counts in the peritoneum and peripheral blood after caecal ligation puncture [8]. In addition, neutrophils isolated from A$_3$AR-knockout mice show impaired migration in response to chemoattractant stimulation [3]; indeed, a growing body of evidence suggests that A$_3$ARs regulate processes that involve cytoskeletal remodelling (for example, chemotaxis) [3,9]. Initial investigations into the cellular localization of neutrophil A$_3$ARs suggested that they are distributed in a highly polarized fashion on the cell membrane [3]. However, until recently, the only widely available methods for investigating the localization of receptors (such as the A$_3$AR) during typical neutrophil behaviours were cell fixation and immunocytochemistry or the overexpression of tagged receptors in model cell lines (for example, differentiated HL60 cells). Both techniques have severe limitations; the former does not permit imaging of living cells, while the latter is a relatively artificial model of primary human neutrophils.

Here, we have further investigated the localization and role of endogenous A$_3$ARs in living human neutrophils using the recently developed fluorescent A$_3$AR ligand CA200645 [10]. Using live-cell imaging, we found that endogenous A$_3$ARs aggregate into plaque-like microdomains on human neutrophil membranes. In addition to facilitating cell migration, A$_3$AR plaques were associated with membrane projections that could extend many times the lengths of the cells. Similar structures, termed 'tethers',

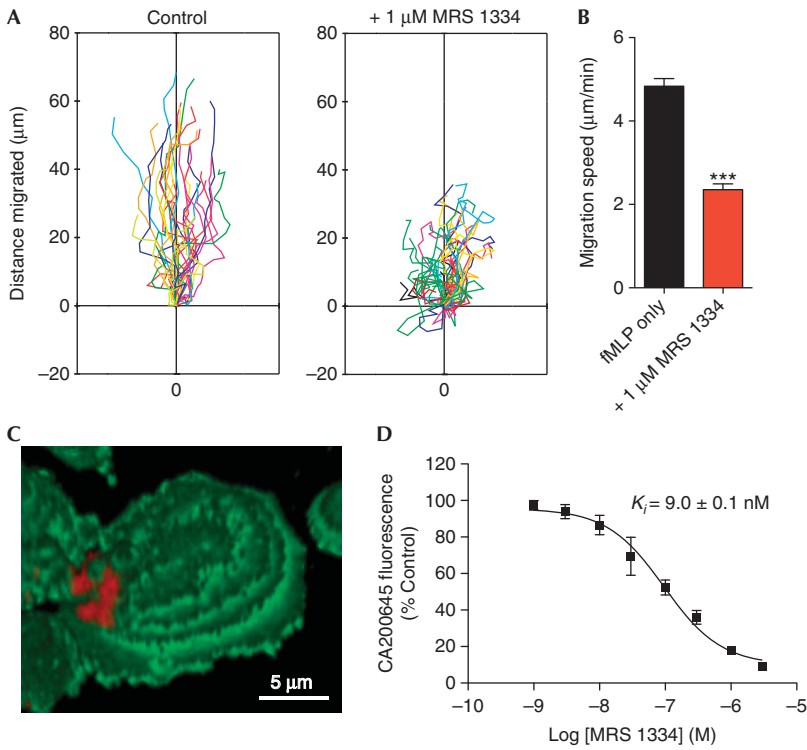

Fig 1 | Endogenous A$_3$ARs aggregate into plaque-like microdomains that regulate processes involving cytoskeletal remodelling. (**A**) Representative plots showing directional migration of neutrophils within fMLP gradients is inhibited by A$_3$AR-selective antagonist MRS1334. (**B**) Quantification showing 51% reduction (***$P < 0.0001$; Student's $t$-test) of migration speed of MRS1334-treated cells ($n = 97$–$127$ cells, three separate experiments). (**C,D**) Flow cytometry-based displacement experiments using CA200645 and unlabelled MRS1334 showed that the fluorescent ligand predominantly detected A$_3$ARs. Data shown are means ± s.e.m. of three separate experiments. A$_3$AR, A$_3$-adenosine receptor; fMLP, fMet-Leu-Phe.

'tubovascular extensions' or 'cytonemes', can be observed using scanning electron microscopy and have been implicated in neutrophil adherence, migration, interactions with bacteria and the phagocytosis of labelled beads [11–14]. Here, we demonstrate the involvement of A$_3$ARs in the formation of these structures, which enable neutrophils to sample, capture and 'reel in' pathogens for phagocytosis.

## RESULTS AND DISCUSSION
### A$_3$ARs aggregate in plaque-like microdomains
In keeping with previous findings regarding the role of A$_3$ARs in neutrophil migration, the highly selective A$_3$AR antagonist MRS1334 [15], at a concentration of 1 μM, inhibited the migration of neutrophils in response to fMLP in a microscope-based migration assay (Fig 1A), reducing the average migration speed by approximately half (Fig 1B). We investigated the localization and role of endogenous A$_3$ARs in living human neutrophils using the recently developed fluorescent A$_3$AR ligand CA200645, a xanthine amine congener derivative connected via a linker to a BODIPY 630/650 fluorophore [10]. Confocal imaging of neutrophils incubated with CA200645 and the general membrane stain Vybrant DiO revealed that A$_3$ARs aggregate into highly polarized, plaque-like microdomains (Fig 1C). As this polarized distribution made it difficult to assess the specificity of CA200645 binding by microscopy, we used flow cytometry to assess the total binding of CA200645 and investigate the

pharmacology of these microdomains in live cells. Competitive binding experiments revealed that MRS1334 blocked the binding of CA200645 with a $K_i$ value of $9.0 \pm 0.1$ nM ($n = 4$; Fig 1D). Given the ~100,000-fold selectivity of MRS1334 for the A$_3$AR compared with other adenosine receptor subtypes and the similarity of this value to the published $K_i$ of MRS1334 at the human A$_3$AR (2.7 nM) [16], we concluded that CA200645 predominantly detects A$_3$ARs on human neutrophils.

### A$_3$ARs promote the formation of membrane protrusions
Neutrophil A$_3$AR plaques were often associated with long, thin membrane protrusions that were typically < 500 nm in diameter and frequently exceeded 10 μm in length (Fig 2A). Analysis of images collected using neutrophils isolated from three healthy volunteers revealed that nearly 70% ($69.3 \pm 6.7\%$) of these protrusions were associated with A$_3$AR plaques. Structures with similar characteristics in neutrophils have been given a number of designations, including 'tethers' and 'cytonemes' [11,13]. To determine whether A$_3$ARs were involved in the formation of these structures, we used the potent, A$_3$AR-selective agonist 2-chloro-N$^6$-(3-iodobenzyl)adenosine-5′-N-methylcarboxamide (2-Cl-IB-MECA), which exhibits 1,400-fold selectivity for A$_3$ over A$_{2a}$ adenosine receptors [15]. We found that treatment with 2-Cl-IB-MECA (1 μM) promoted the rapid extension of these structures (Fig 3A). To quantify the tether-promoting effects of A$_3$AR activation, we performed a series of experiments in which

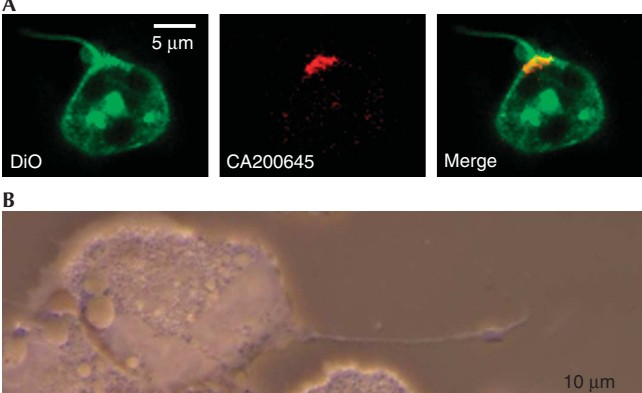

**Fig 2** | A₃AR activation promotes the extension of membrane projections from human neutrophils. (**A**) Confocal imaging of Vybrant DiO/CA200645-labelled human neutrophils revealed that A₃AR plaques were frequently associated with membrane projections. Quantitative analysis of neutrophils isolated from the blood of three different healthy volunteers revealed that $69.3 \pm 6.7\%$ of these projections were associated with A₃AR plaques. (**B**) A Riveal Contrast Microscope (Quorum, Ontario, Canada) was used to show that these projections are also found on unstained, phorbol 12-myristate 13-acetate-treated human neutrophils. A₃AR, A₃-adenosine receptor.

we treated neutrophils with either Hank's Buffered Salt Solution (HBSS) or 2-Cl-IB-MECA, in the presence or absence of MRS1334 ($1\,\mu M$) or the actin polymerization inhibitor cytochalasin B ($300\,nM$). The data from these experiments were randomized, and the number of cytonemes per cell before and 7.5 min following HBSS or 2-Cl-IB-MECA treatment was determined by a blind observer using predefined criteria (described in the Methods section). The results of 34–35 individual experiments using neutrophils isolated from the blood of nine healthy volunteers (559–674 cells total) revealed that 2-Cl-IB-MECA promoted an increase in the percentage of cells exhibiting cytonemes (Fig 3B) and the number of cytonemes/cell (3C), effects that were blocked by MRS1334 (Fig 3D). Cytochalasin B also inhibited 2-Cl-IB-MECA-induced cytoneme formation (Fig 3C). Imaging of differentiated HL60 cells (which mimic neutrophil behaviour and show both plaques and cytoneme projections) expressing the filamentous-actin probe LifeAct revealed that these structures were composed of F-actin (Fig 3D). Neither MRS1334 nor cytochalasin B significantly affected cytoneme formation, as can be seen from the pre-agonist treatment bars in Fig 3C. Imaging experiments performed in the presence of the DNA stain SYTO 83 revealed that cytonemes were not associated with DNA-based neutrophil extracellular traps (supplementary Fig S1 online). Thus, neutrophil cytonemes are actin-based structures that protrude from A₃AR-rich plaques whose formation can be stimulated in an A₃AR-dependent manner.

## Cytonemes enable neutrophils to capture pathogens

Filipodia and nanometer-thin micropodial extensions (NMEs) have been shown to enable the capture and 'reeling in' of beads and pathogens in macrophages [17] and epithelial cells [18], respectively. In the case of NMEs, the capture of bacteria has a critical role in epithelial invasion and is driven by extracellular ATP [18]. Although cytonemes differ from NMEs (particularly in terms of maximum length), previous studies using scanning electron microscopy indicated that they might have a similar role in neutrophils [12]. We postulated that A₃AR-induced cytonemes could represent a mechanism for the targeted, long-range capture or sampling of microbial pathogens. To test this hypothesis, we exposed human neutrophils to SYTO 83-stained *Escherichia coli*. Although some cells exhibited typical chemotactic or phagocytic behaviours, a number of cells rapidly extended cytonemes in response to bacterial exposure (supplementary Videos S1 and S2 online; green, neutrophils; orange, *E. coli*). The images shown in Fig 4A depict two cells rapidly extending cytonemes, capturing SYTO 83-stained *E. coli* and retracting the captured bacteria back to the cell surface. CA200645 labelling showed that these bacteria-tethering structures were associated with the A₃AR plaques (Fig 4B). Quantitative experiments revealed that *E. coli* exposure approximately doubled cytoneme number (Fig 4C). This increase was suppressed by MRS1334, indicating a role for the A₃AR in bacteria-mediated cytoneme extension.

## Cytonemes enable long-range capture of pathogens

To further investigate the physical properties of neutrophil cytonemes and confirm that this behaviour was not restricted to one type of bacteria, experiments were performed using SYTO 83-stained *Helicobacter pylori*. As these bacteria can adhere to fibronectin-coated substrates [19], thus slowing/halting the cytoneme retraction process, they were a useful model for investigating the physical properties of these transient structures. As shown in Fig 5A, these experiments revealed that cytonemes enable the capture/sampling of pathogens up to $100\,\mu m$ away from the cell. We also found that cytonemes were capable of tethering Gram-positive bacteria, such as *Staphylococcus aureus* (supplementary Fig S2 online). Neutrophils are capable of forming several cytonemes at a time, which can be used to contact several bacteria (Fig 5B). Notably, the images shown in Fig 5B suggest that the capture of pathogens by cytonemes is an active, rather than passive, process. The images depict a neutrophil extending a cytoneme towards a moving *H. pylori* cluster (indicated by the white arrows); over the course of the time series, the orientation of the cytoneme changes several times before tethering the bacterial cluster. Soon thereafter, tension simultaneously builds in all of the cytonemes that have been extended by the neutrophil, presumably as it attempts to 'reel in' the fibronectin-bound bacteria (supplementary Video S3 online). Previous work using epithelial cells and carboxylated beads found that the retraction of individual filipodia is dependent on receptor–ligand interactions at the filipodial tip [20]; interestingly, however, the simultaneous exertion of force suggests that all of the cytonemes extended by a single neutrophil might be controlled by a common motor/sensory complex. Finally, we tested whether A₃ARs could be targeted to enhance the phagocytic capacity of neutrophils using pHrodo *S. aureus*-labelled bioparticles, finding that 2-Cl-IB-MECA stimulation enhanced the ability of neutrophils to phagocytose the bioparticles (Fig 5C).

The findings presented here not only highlight a previously unknown function of A₃ARs, but also shed extra light on an

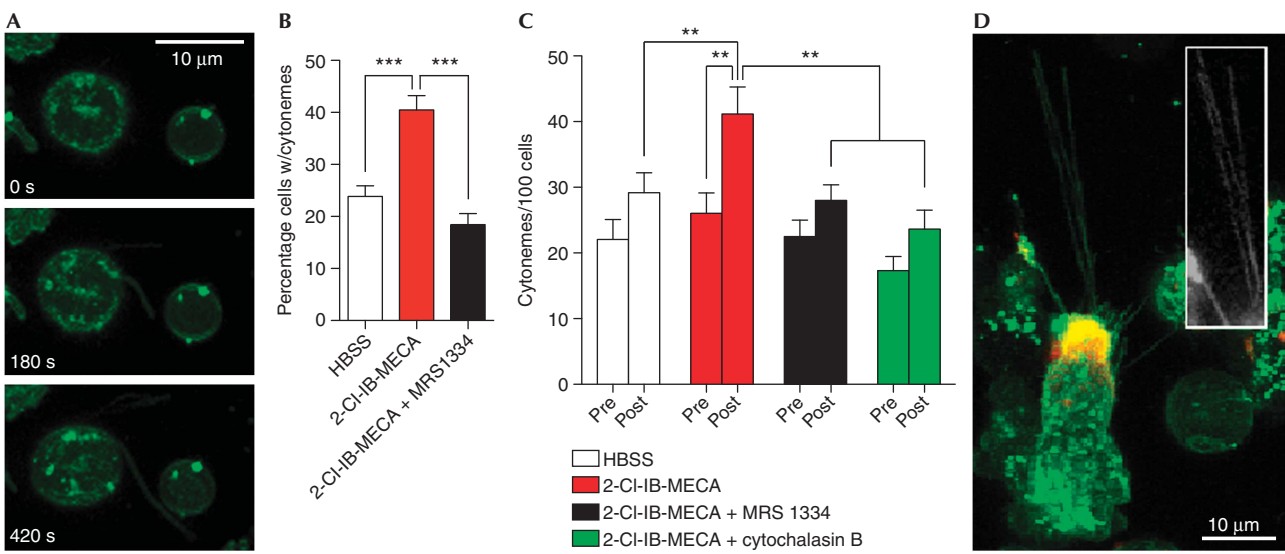

**Fig 3** | The A$_3$AR-selective antagonist MRS1334 and cytochalasin B inhibit A$_3$AR agonist-stimulated cytoneme formation. (**A**) The A$_3$AR-selective agonist 2-Cl-IB-MECA (1 μM at $t = 0$ s) promotes the extension of cytonemes from neutrophils. Quantitative analysis revealed that 2-Cl-IB-MECA treatment significantly increases the percentage of cells exhibiting cytonemes (**B**) and the number of cytonemes/cell (**C**), effects inhibited by MRS1334 or cytochalasin B. Data shown are means ± s.e.m. of 26–35 individual experiments; significance was determined via one-way analysis of variance with *post-hoc* Newman–Keuls tests. (**D**) ATRA-differentiated HL60 cells, which exhibited both A$_3$AR plaques and plaque-associated cytonemes (merged image of a Vybrant DiO and CA200645-labelled HL60 cell), were transfected with the fluorescent F-actin probe LifeAct, confirming the presence of F-actin in these structures (inlay, white). **$P < 0.01$, ***$P < 0.001$. A$_3$AR, A$_3$-adenosine receptor; HBSS, Hank's Buffered Salt Solution; 2-Cl-IB-MECA, 2-chloro-N$^6$-(3-iodobenzyl)adenosine-5'-N-methylcarboxamide.

underappreciated and poorly understood behaviour of neutrophils (that is, cytoneme-mediated capture of bacteria) [11,12]. It has become increasingly apparent, for example, with the discovery of DNA-based traps [21], that neutrophils can deploy different strategies for the sampling, capture and clearance of invading pathogens. Our results provide further evidence that cytonemes could enable the targeted phagocytosis of pathogens that might otherwise be difficult to access. It is worth noting that recent studies of nanotubes, actin-based membrane protrusions that enable the long-range transport of signals and organelles between cells, have found that Connexin 43 hemichannels are localized at the base of these protrusions, much like the case of A$_3$ARs presented here [22]. Connexin hemichannels have been shown to facilitate the release of ATP from many cell types [23], and in epithelial cells, ATP released from Connexin 26 hemichannels have been demonstrated to facilitate bacterial invasion and spreading [24]. In a related study, Romero *et al* [18] recently showed that extracellular ATP stimulates bacterial capture by epithelial cells via NMEs. In the case of neutrophils, Connexin 43 hemichannels have been shown to facilitate the release of ATP [25], which is rapidly converted to adenosine by neutrophil ecto-ATPases [7], suggesting the possible existence of an autocrine signalling pathway for the formation of cytonemes under conditions requiring a heightened surveillance state.

Our findings raise more questions for further study; for example, it would be particularly important to investigate whether different pathogens show a range of potencies for inducing cytonemes or possess mechanisms by which to avoid capture by these structures. However, taken alone, the identification of the A$_3$AR as a mediator of both neutrophil chemotaxis and of cytoneme formation and function makes the receptor a potentially important therapeutic target by which to modulate neutrophil function in immunological and inflammatory disorders or infectious diseases.

## METHODS

**Materials.** MRS1334 and 2-Cl-IB-MECA were obtained from Tocris Biosciences. CA200645 was obtained from CellAura Technologies (Nottingham, UK). Vybrant DiO and SYTO 83 were obtained from Life Technologies (Carlsbad, CA). Cytochalasin B, fMLP and HBSS were obtained from Sigma (St. Louis, MO). E. coli (XL1-Blue) were obtained from Agilent Technologies (Santa Clara, CA), and H. pylori (J99)/S. aureus were provided by Dr Jafar Mahdavi.

**Neutrophil isolation.** Human venous blood was collected from healthy volunteers and placed into tubes with EDTA. Neutrophils were isolated using Polymorphprep (Axis-Shield, Dundee, Scotland) according to the manufacturer's protocol. Blood was drawn according to a protocol approved by the local ethics committee.

**Preparation of fibronectin-coated slides.** Sterile, glass-bottomed culture dishes (MatTek, Ashland, MA) and chambered coverglass slides (Nalge Nunc International, Rochester, NY) were coated with fibronectin (10 μg/ml in HBSS, Sigma). Slides/dishes were incubated at room temperature (RT) for 30 min and washed three times with HBSS before use.

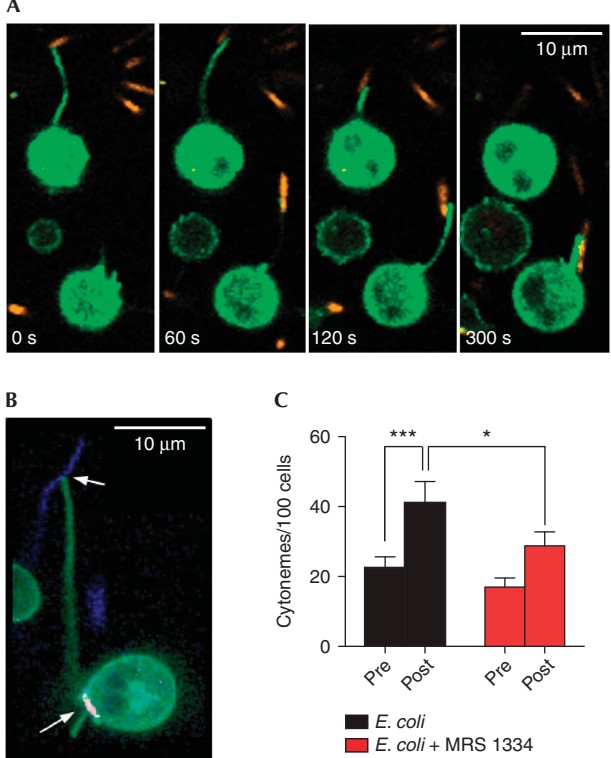

**Fig 4** | Neutrophil cytonemes formed in response to E. coli exposure are capable of tethering and 'reeling in' bacteria for phagocytosis. (**A**) Time-lapse confocal imaging shows neutrophils (Vybrant DiO labelled; green) can capture E. coli (SYTO 83 labelled; orange) via extension of cytonemes, which are rapidly retracted for phagocytosis. (**B**) Imaging of CA200645-labelled neutrophils showed bacteria-tethering cytonemes are associated with A₃AR plaques (red); SYTO 83-labelled E. coli are pseudocoloured blue. (**C**) Exposure of neutrophils to E. coli promotes cytoneme extension, an effect inhibited by MRS1334 ($n = 26$ individual experiments). Data shown are means ± s.e.m.; statistical significance was determined via repeated measures analysis of variance with *post-hoc* Newman–Keuls tests. $*P < 0.05$, $***P < 0.001$. See also supplementary Videos S1, S2 online. A₃AR, A3-adenosine receptor.

**Microscope-based migration assays.** Neutrophils ($2.5 \times 10^6$ cells/ml) were added to fibronectin-coated 35-mm glass-bottomed dishes and incubated for 10 min at 37 °C to allow cells to settle. The dishes were then placed into a temperature-controlled stage (maintained at 37 °C) on an Axiovert S100 microscope (Carl Zeiss GmbH, Jena, Germany). Sterile Femtotips were loaded with 100 nM fMLP and connected to a FemtoJet system (Eppendorf, Hamburg, Germany). Cells were visualized using a 40 × /0.60 NA objective, and an Eppendorf Micromanipulator 5,171 was used to position the Femtotips. Continuous fMLP release was achieved by setting the compensation pressure to 100 hPa; and time-lapse images were captured using Axiovision software (v4.8; Carl Zeiss GmbH). Migration paths were traced using ImageJ (v1.44p; NIH, Bethesda, MD) with Manual Tracking (F. Cordeli, France).

**Flow cytometry.** Neutrophils were incubated with and without MRS1334 for 20 min at RT; after an additional 20 min at RT with 30 nM CA200645, cells were analysed using a Coulter FC500 flow cytometer (Beckman Coulter, Brea, CA). Median CA200645 fluorescence intensity was determined Using WinMDI software (v2.8; J. Trotter, Scripps Research Institute, La Jolla, CA). A competitive binding curve was generated using Prism (v.4.0; GraphPad Software, San Diego, CA).

**Confocal imaging.** All images shown were captured using a Zeiss LSM710 microscope (Carl Zeiss GmbH) with a × 63 plan-Apochromat/1.40NA oil DIC objective. Lasers (488 nm argon, 561 nm DPSS and 633 nm HeNe) were used to excite Vybrant DiO, SYTO 83 and CA200645. Neutrophils were labelled with Vybrant DiO according to the manufacturer's protocol. Roughly $1 \times 10^6$ labelled neutrophils were suspended in HBSS and added to a fibronectin-coated, chambered coverglass slide, which was maintained at 37 °C. Zen 2010 software (v6.0; Carl Zeiss GmbH) was used to capture image stacks, with image slices captured at 0.5 μm intervals. The image stacks were rendered as maximum intensity projections or three-dimensional images for analysis/presentation. For experiments that required the visualization of A₃ARs, an additional 20-min incubation at 37 °C with CA200645 was performed before DiO staining. Bacteria were stained with SYTO 83 according to the manufacturer's protocol.

**HL60 differentiation and transfection.** HL60 cells were differentiated with 1 μM all-*trans* retinoid acid (Sigma); after 108 h, the cells were transfected with the LifeAct construct (Ibidi GmbH, Munich, Germany) using a Nucleofector system (Lonza, Basel, Switzerland). After 12 h, the HL60 cells were labelled with Vybrant DiO and CA200645, and actin distribution was observed.

**Cytoneme quantification assays.** Neutrophils were pre-incubated in HBSS alone or with MRS1334 (1 μM final) or cytochalasin B (300 nM final) for 20 min or 1 h, respectively. HBSS, 2-Cl-IB-MECA (1 μM final) or *E. coli* ($2.5 \times 10^7$ cells/ml) were added 2.5 min after image capture was initiated. To avoid subjective bias, data were randomized and analysed by a researcher not involved in performing the experiments. Contrast and gain were adjusted to aid in visualization of the cytonemes, and the number per cell immediately before and 7.5 min after the described treatments were counted. Cytonemes were only included if they originated from a viable cell, were <1 μm in diameter and more than 5 μm long, and exhibited some movement (on the basis of time series analysis). Cells were only included if 50% or more of the cell body was visible. Structures that originated at the cell base (that is, on the coverslip) were excluded. Three experiments were excluded owing to red blood cell contamination (which made it difficult to determine cytoneme numbers and potentially altered the signalling environment). One statistical outlier that fell more than two s.d.'s outside the mean was removed.

**Phagocytosis assay.** Human neutrophils were combined in an uncoated 96-well plate with pHrodo Red S. aureus Bioparticles (Life Technologies) as specified by the manufacturer. Plates were incubated at 37 °C, and phagocytosis was assessed by measuring fluorescence intensity at 15 min intervals using a SpectraMax plate reader (Molecular Devices) at excitation and emission wavelengths of 560 and 585 nm, respectively.

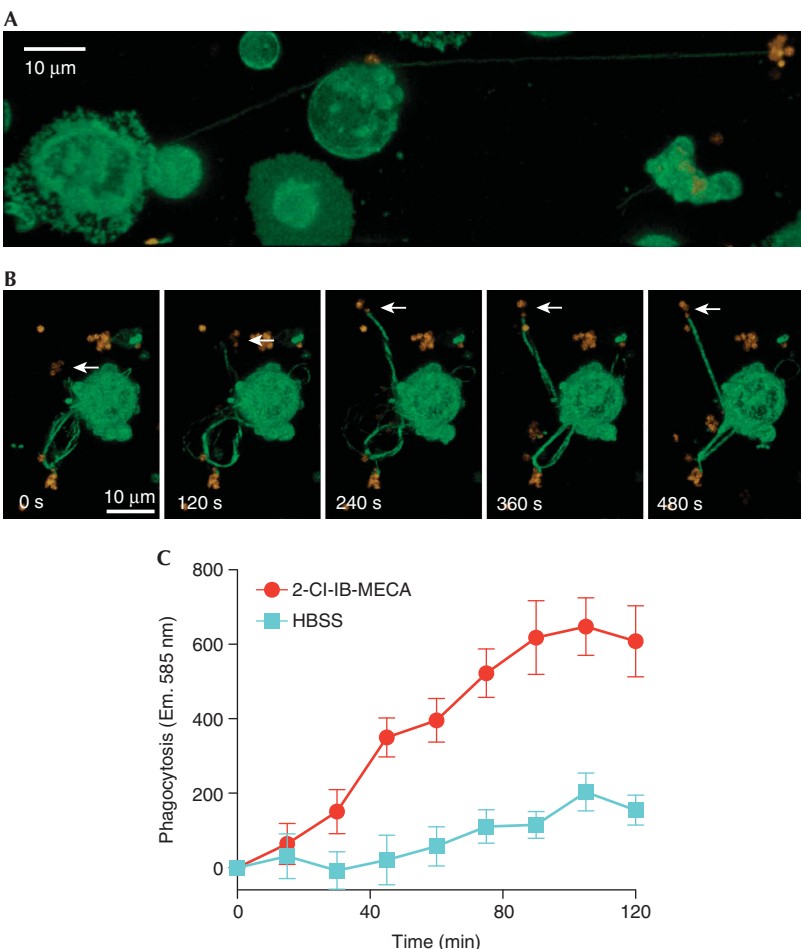

**Fig 5 | Neutrophil nanotubes enable the targeted, long-distance sampling/capture of pathogens. (A)** Confocal imaging of Vybrant DiO-stained neutrophils that were exposed to SYTO 83-stained H. pylori revealed that cytonemes enable neutrophils to sample/capture bacteria up to 100 µm away from the cell. In addition, time-lapse imaging revealed that neutrophils are capable of simultaneously extending several cytonemes and capturing several pathogens at a time. **(B)** The spatial adaptation of cytonemes in relation to the position of motile bacteria (for example, the H. pylori cluster highlighted with white arrows) suggests that cytonemes are capable of adapting to changing spatio-temporal signals. See also supplementary Video S3 online. **(C)** Activation of A$_3$ARs with 2-Cl-IB-MECA enhanced the ability of human neutrophils to phagocytose fluorescent S. aureus bioparticles. Data shown are means ± s.e.m. of three individual experiments. Y axis shows relative intensity units. A$_3$AR, A3-adenosine receptor; HBSS, Hank's Buffered Salt Solution; 2-Cl-IB-MECA, 2-chloro-N$^6$-(3-iodobenzyl)adenosine-5′-N-methylcarboxamide.

**Statistical analysis.** All of the statistical analyses described in the figure legends were performed using Prism v4.0.

ACKNOWLEDGEMENTS
This work was supported by the Medical Research Council (grant number G0800006; S.J.H., S.J.B. and B.K.) and NIH/GLRCE U54 (grant number AI057153; V.N.).

*Author contributions*: R.C., S.J.B. and S.J.H. designed experiments and analysed data. R.C. performed all of the experimental work, except the Riveal Contrast experiments (performed by K.A.-M.). T.S. analysed and assisted with imaging. V.N. gave conceptual advice. B.K., S.J.B. and S.J.H. conceived and managed the project. All authors contributed to manuscript preparation.

CONFLICT OF INTEREST
S.J.H. and B.K. are founders and directors of the University of Nottingham spin-out company that provided CA20065.

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
