## [Review Process File · EMBO Reports]

Manuscript EMBOR-2012-36758

Adenosine-A3 receptors in neutrophil microdomains promote the formation of bacteria-tethering cytonemes

Ross Corriden, Tim Self, Kathryn Akong-Moore, Victor Nizet, Barrie Kellam, Stephen J. Briddon and Stephen J. Hill

Corresponding author: Stephen J. Hill, University of Nottingham

Review timeline:	Submission date:	23 October 2012
	Editorial Decision:	05 December 2012
	Revision received:	06 May 2013
	Editorial Decision:	24 May 2013
	Revision received:	04 June 2013
	Accepted:	06 June 2013

Editors: Alejandra Clark, Nonia Pariente

Transaction Report:

1st Editorial Decision

05 December 2012

Thank you for the submission of your research manuscript to EMBO reports. We have now received the full set of reports on it.

As you will see, although the referees find the study potentially interesting they raise a number of significant concerns, most of which will need to be experimentally addressed to make the study fully conclusive. One important concern raised by referee #2 refers to the nature and identity of the A3AR induced filopodia-like extensions. This referee suggests that additional functional characterisation of these structures is needed to label them as nanotubes. In addition he/she is concerned about the novelty of the findings indicating that important citations of previous work are currently missing which must be added and discussed. All referees indicate that a number of essential controls, including quantification of results must be included. Referees #1 and #3 suggest that in order to conclusively demonstrate that A3AR activity promotes the formation of filopodia-like extensions in neutrophils, A3AR knock-out neutrophils used in previous studies should be examined.

Given the potential interest of the findings and considering that all referees provide constructive suggestions on how to move the study forward, I would like to give you the opportunity to revise the manuscript, with the understanding that all referees concerns, including those points mentioned

above have to be addressed and that acceptance of the manuscript would entail a second round of review. I would like to point out that it is EMBO reports policy to allow a single round of revision and thus, acceptance or rejection of the manuscript will depend on the outcome of the next final round of peer-review.

Revised manuscripts should be submitted within three months of a request for revision; they will otherwise be treated as new submissions. If you feel that this period is insufficient for a successful submission of your revised manuscript I can potentially extend this period slightly. Also, the length of the revised manuscript should not exceed roughly 30,000 characters (including spaces). Should you find the length constraints to be a problem, you may consider including some peripheral data in the form of Supplementary information. However, materials and methods essential for the repetition of the key experiments should be described in the main body of the text and may not be displayed as supplemental information only.

As part of the EMBO publication's Transparent Editorial Process, EMBO reports publishes online a Review Process File to accompany accepted manuscripts. This File will be published in conjunction with your paper and will include the referee reports, your point-by-point response and all pertinent correspondence relating to the manuscript.

We also welcome the submission of cover suggestions or motifs that might be used by our Graphics Illustrator in designing a cover.

I look forward to seeing a revised form of your manuscript when it is ready. Should you in the meantime have any questions, please do not hesitate to contact me.

REFEREE REPORTS:

Referee #1

In this manuscript Corriden and co-workers examine the involvement of the A3-Adenosine receptor in the formation of nanotube extensions from neutrophils, then describe and characterize the functions of those nanotubes.

The major concerns I have with this manuscript are that additional controls should be run for certain aspects of the study and additional evidence should be presented to solidify the conclusions reached by the authors.

Specific Comments

Although agonists and antagonists were used in this manuscript, whenever pharmacological agents are used there are always the possibilities of non-specific effects. A3AR knock-out mice exist and two of the authors have previously used neutrophils from those animals (pg 3-4). What are the effects of using neutrophils from those knock-out mice. Is it in-line with the evidence presented in the manuscript. Do neutrophils that are devoid of A3AR cease to form nanotubes? If they do form nanotubes can they capture microbes and function properly?

The localization seen in Fig 1C must be controlled for. Does a different derivative connected to a linker and BODIPY show the same or different localization patterns? Does the linker attached to BODIPY alone localize to patches on the neutrophil?

In figure 3B, a MRS1344 only and a cytochalasin-B only graph should be included for comparison.

The authors claim through the use of cytochalasin B that the nanotubes are made of actin. This is

evidence alone is not sufficient for that claim. Filamentous actin should be stained for and examined in the nanotubes.

The authors show that nanotubes reach-out to grab *E. coli* and *H. Pylori*. To confirm that these structures target bacteria specifically additional controls are needed. Will these structures grab beads of the same size or will they grab anything nearby? Is size important? Also do they function in a similar manner with Gram+ bacteria?

Minor comments

Scale bars are needed on all images.

Referee #2

In this article, the authors identified the A₃-adenoside receptor (A₃ AR) as an important regulator of neutrophil migration using a new fluorescent A₃ AR ligand. In addition to its effects on cell migration, the authors observed the development of long filopodia-like extensions that were dependent on A₃ AR activation. These extensions that the authors name "nanotubes" enabled the cells to capture pathogens for phagocytosis.

General comments:

As noted by the authors on p4., similar structures have been described in neutrophils, including their interactions with bacteria. In addition, this type of capture and "reeling in" of pathogens as also been shown with macrophages (Vonna et al. *European Biophysics J.*, 2006) and with epithelial cells (Romero et al. *Cell Host & Microbe*, 2011), which was not mentioned by the authors. In addition, the authors also talk about the "simultaneous exertion of force" p8 observed in their system when the neutrophils attempt to "reel in" the pathogens. However, they failed to mention a recently published article, that specifically looks at the forces exerted by filopodia binding to beads upon retraction toward the cell (Romero et al. *J. Cell Sci.*, 2012). Overall, the authors need to better describe what was already known in the field prior to their study in order to clarify what was new in their study.

The second main general concern is the choice of the word "nanotube" to describe the structure they observed. As the authors correctly state on p. 6, when talking about neutrophils, similar structures have been named "tethers" and "cytonemes". They incorrectly state however that the same structures have been called nanotubes in other cell types. There are distinctions between the various structures and their names. Nanotubes

imply that these tubular structures are able to transfer various materials (ie. Organelle transfer, Ca flux...) from one cell to another. Tethers or cytonemes can bridge two cells but do not transfer cytosolic materials. Thus, the authors can't simply choose to call their structures nanotubes based on their physical appearance. They must look at their function, and unless they can show the transfer of material they can't call them nanotubes.

Specific comments:

Fig. 1: Nice figure. The authors show the role of A₃ AR in cell migration. They show nice aggregation of A₃ AR in microdomains and show the specificity of CA200645 binding to A₃ AR.

Fig. 2 & 3: I am not sure why these two figures are not combined into 1.

Fig. 2: simply show neutrophils being able to form filopodia extension (which is already known). The only new data is Fig. 2A which show CA200645 at the base of the filopodia. One picture without any quantitation is not very telling. The authors should quantify how often they observed CA200645 at the base of filopodia.

Fig. 4: p7. The authors postulated that "nanotubes" could "represent a mechanism for the targeted, long-range capture". They must add something regarding the similarities or differences with previous data on nanometer-thin micropodial extensions (NMEs) (Romero et al. *Cell Host & Microbe*, 2011).

Fig. 5.B: see general comment on forces.

Overall, the authors do show that A₃AR is important for the formation and function of these filopodia-like structures leading to the capture of bacteria. However, as is, I do not believe that the authors showed enough to warrant publication in this journal.

Referee #3

In this manuscript by Corriden et al, the authors use both qualitative and quantitative microscopic measures to evaluate the role of the neutrophil A₃-Adenosine Receptor in promoting neutrophil nanotube formation. The authors then expand their study by linking A₃AR-dependent nanotube formation to the ability of neutrophils to sense, tether, and reel in bacterial pathogens. In recent years, the A₃AR receptor has been shown to be important for neutrophil chemotaxis as influenced by ATP release. Additionally, the ability of neutrophils to elaborate long tubular extensions (spanning up to 100µM in length) capable of contacting/sensing bacterial pathogens has also been previously described for *Salmonella enterica* serovar Typhimurium. This manuscript elaborates on these prior studies by establishing a direct association between A₃AR cellular localization as well as A₃AR signaling and the enhanced production of neutrophil nanotubes capable of sensing bacteria. A₃AR mediated enhancement of nanotube formation implies a novel mechanism by which neutrophils can sense and respond to bacterial insult. In this reviewer's opinion, the work described herein is both novel and appealing. While the majority of the experimental design is sound, there are a number of points that should be addressed prior to publication.

1. Can the authors cite or provide evidence for both the potency and selectivity of the agonist 2-Cl-IB-MECA? Because the authors draw important conclusions from the use of this agonist it seems essential to at least describe in some detail its degree of selectivity in the text.

2. On page 7, the authors state "a number of cells rapidly extended nanotubes in response to bacterial exposure". Can the authors provide a quantitative assessment of the percentage of neutrophils that elaborated nanotubes in response to bacterial exposure (i.e. are only a small subset of total neutrophils responding in this manner)? This is particularly important because in figures 3 and 4 the authors present quantitative data as nanotubes/100 cells. Unfortunately, quantifying in this manner does not give a precise indication of how many neutrophils are actually extending nanotubes (as the authors state, a single neutrophil is capable of elaborating multiple nanotubes). The data as presented in figures 3 and 4 would be acceptable only if the authors first provide a percentage of neutrophils extending nanotubes (independent of how many nanotubes are actually present on a single cell) compared to cells not extending nanotubes.

3. In figure 3B, this reviewer wonders what percentage of nanotubes actually co-localize with A₃AR both before and after 2-Cl-IB-MECA treatment. For example are the 20-30 nanotubes/100cells seen before agonist treatment independent of A₃AR co-localization? If these "pretreatment nanotubes (no agonist)" are indeed associated with A₃AR co-localization it is more difficult to directly associate nanotube formation with an A₃AR function, in particular because MRS1334 treatment does not reduce nanotube formation below "pretreatment" levels. See #4 for a potential means by which the authors could experimentally address this point.

4. In a previous study of R. Corriden (Ref. 3 in this manuscript), neutrophils from an A₃AR KO mouse are used to evaluate the contribution of the A₃AR receptor to neutrophil chemotaxis in response to ATP release. Are these mice accessible? If so, this reviewer feels it would be extremely valuable to assess nanotube formation using WT and KO murine neutrophils. This would help to shed light on to a number of ambiguities that exist in the manuscript regarding how often and in what capacity A₃AR is associated with neutrophil nanotubes.

General Comments

We wish to thank each of your reviewers for their helpful suggestions and feedback. Two of the reviewers suggested that if available, experiments using neutrophils isolated from A3AR knock-out animals would potentially be informative. As noted by the reviewers, two of the authors on this manuscript (Ross Corriden and Victor Nizet) are co-authors in a study that utilized these mice; however, the colony was maintained by a collaborator and no longer exists. Furthermore, the laboratory in which most of this work took place (the Hill laboratory at the University of Nottingham) does not work with animals and thus does not have the necessary space/approval to do such experiments. Nevertheless, Ross worked with the Nizet laboratory at the University of California, San Diego to obtain the A3AR knock-out mice from Merck, where they were originally generated. Unfortunately, the investigator who generated the line recently left the company, slowing progress on a material transfer agreement that, at the time of writing, is still under negotiation.

Furthermore, isolated mouse neutrophils are known to exhibit differences in behavior compared to human neutrophils. For example, although 80% of human neutrophils project neutrophil extracellular traps (NETs) in 3-4 hours following pro-inflammatory stimulation, only 30% of mouse neutrophils do so after 16 hours. Differences such as these would likely make it quite difficult to obtain a meaningful amount of data relating to filipodia-like structures using currently available methods.

Reponses to Specific Comments

Referee #1

Specific Comments

Although agonists and antagonists were used in this manuscript, whenever pharmacological agents are used there are always the possibilities of non-specific effects. A3AR knock-out mice exist and two of the authors have previously used neutrophils from those animals (pg 3-4). What are the effects of using neutrophils from those knock-out mice. Is it in-line with the evidence presented in the manuscript. Do neutrophils that are devoid of A3AR cease to form nanotubes? If they do form nanotubes can they capture microbes and function properly?

Unfortunately, we did not have access to a colony of A3AR knock-out mice at the time this manuscript was written and were not able to obtain A3AR knock-out mice in sufficient time to perform additional experiments for the manuscript (please see our explanation above). However, we believe we were rigorous in our selection and use of highly selective agonist/antagonists of the A3AR receptor, using concentrations that could reasonably be expected to not lead to non-specific effects.

The localization seen in Fig 1C must be controlled for. Does a different derivative connected to a linker and BODIPY show the same or different localization patterns? Does the linker attached to BODIPY alone localize to patches on the neutrophil?

Although we do not have easy access to individual components of the fluorescent ligand used in this study, which was not produced in-house, the fluorescent ligand was extensively validated in a recently published manuscript (Stoddart et al., *Chem Biol.* 2012). Although it is possible to use imaging approaches to quantitatively assess binding, we believe that the most reliable, un-biased and quantitative way to assess the specificity of CA200645 binding in a non-adherent cells such as human neutrophils is flow cytometry with an unlabeled and highly selective A3AR antagonist (i.e., MRS1334). Using such an approach, we were able to almost entirely block binding (>90% of total binding; Figure 1D) and eliminate the detected fluorescence using MRS1334, and the K_i value derived from the displacement data is very close to the published value for MRS1334 at the human A3AR.

In figure 3B, a MRS1344 only and a cytochalasin-B only graph should be included for comparison.

The data in Figure 3B were obtained using cells that were pre-incubated with MRS1334 or cytochalasin-B for 20 min or 1 h, respectively. Thus, although it was not clearly presented in the text, the effects of MRS1334 and cytochalasin-B alone can be determined from the “pre” treatment bars. We have adjusted the text to make this clearer.

The authors claim through the use of cytochalasin B that the nanotubes are made of actin. This is evidence alone is not sufficient for that claim. Filamentous actin should be stained for and examined in the nanotubes.

This is a valid criticism that we have addressed in our revisions. Because fluorescent phalloidin can interfere with actin polymerization, we instead transfected neutrophil-like differentiated HL60 cells with LifeAct, a fluorescent actin binding protein that enables visualization of filamentous actin in live cells without interfering with polymerization. In addition to finding that differentiated HL60s form cytonemes, and that these cytonemes are associated with A3AR plaques, we observed the presence of f-actin in the tethers. This new data has been incorporated into Figure 4 (D).

The authors show that nanotubes reach-out to grab E. coli and H. Pylori. To confirm that these structures target bacteria specifically additional controls are needed. Will these structures grab beads of the same size or will they grab anything nearby? Is size important? Also do they function in a similar manner with Gram+ bacteria?

These are important questions that we have attempted to address with our revisions. Thus far, we have not found evidence that the tethers are capable of grabbing anything other than beads/bacteria. Similar experiments with the Gram-positive bacterium *Staphylococcus aureus* revealed that these bacteria are also targeted by membrane tethers; a representative image from these experiments have been added as Supplemental Figure 2. We also performed experiments with fluorescent *Staphylococcus aureus*-labeled beads to obtain more easily quantifiable functional data. In these experiments, isolated human neutrophils were mixed with the labeled beads in 96 well plates that were deliberately left uncoated to limit neutrophil motility. This new data revealed that stimulation with the A3AR agonist 2-Cl-IB-MECA strongly induced phagocytosis of the beads (new Figure 5C).

Minor comments

Scale bars are needed on all images.

Scale bars have been added to all images.

Referee #2

General comments:

As noted by the authors on p4., similar structures have been described in neutrophils, including their interactions with bacteria. In addition, this type of capture and "reeling in" of pathogens as also been shown with macrophages (Vonna et al. European Biophysics J. , 2006) and with epithelial cells (Romero et al. Cell Host & Microbe, 2011), which was not mentioned by the authors. In addition, the authors also talk about the "simultaneous exertion of force" p8 observed in their system when the neutrophils attempt to "reel in" the pathogens. However, they failed to mention a recently published article, that specifically looks at the forces exerted by filopodia binding to beads upon retraction toward the cell (Romero et al. J. Cell Sci., 2012). Overall, the authors need to better describe what was already known in the field prior to their study in order to clarify what was new in their study.

We thank the reviewer for their extensive suggestions and apologize for the unintentional omission of this work in our manuscript (at least one of these reports had not yet been published at the time we wrote the initial draft of our manuscript). In our revised manuscript, we have included a number of additional citations, including those you referenced above; furthermore, although our space is somewhat limited given the report-style format of this journal, we have added additional

information regarding key findings in the field (e.g., the ATP-induced capture of bacteria by epithelial cells via nanometer-thin micropodial extensions as described by Romero et al.).

Although the ability of structures similar to the ones we show to capture and reel in pathogens has been elegantly described in macrophages and epithelial cells, the ability of neutrophils to use similar structures to capture pathogens is still relatively poorly understood (as recently as 2011 a paper was published with the title “Strategies of professional phagocytes in vivo: unlike macrophages, neutrophils engulf only surface-associated microbes” [Colucci-Guyon et al., *J Cell Sci*, 2011]). In the context of neutrophil function, we believe that the findings we have presented here using live cells build on the pioneering work of Galkina et al., which was predominantly based on the use of fixed cells and scanning electron microscopy. In addition, using the fluorescent ligand CA200645 to investigate endogenous A3 adenosine receptors in human neutrophils, we found that these receptors aggregate into plaque like microdomains, and activation of these receptors enhances the phagocytic ability of these cells. Taken together, we believe that our data reveal a previously unknown and potentially therapeutically exploitable role of A3 adenosine receptors in host-pathogen interactions.

The second main general concern is the choice of the word "nanotube" to describe the structure they observed. As the authors correctly state on p. 6, when talking about neutrophils, similar structures have been named "tethers" and "cytonemes". They incorrectly state however that the same structures have been called nanotubes in other cell types. There are distinctions between the various structures and their names. Nanotubes imply that these tubular structures are able to transfer various materials (ie. Organelle transfer, Ca flux...) from one cell to another. Tethers or cytonemes can bridge two cells but do not transfer cytosolic materials. Thus, the authors can't simply choose to call their structures nanotubes based on their physical appearance. They must look at their function, and unless they can show the transfer of material they can't call them nanotubes.

We apologize for labeling these structures nanotubes without complete evidence to validate doing so. Based on our current knowledge, and in keeping with the literature, we have instead called these structures “cytonemes”. We have made appropriate changes throughout the text.

Specific comments:

Fig. 1: Nice figure. The authors show the role of A₃ AR in cell migration. They show nice aggregation of A₃ AR in microdomains and show the specificity of CA200645 binding to A₃ AR.

Thank you; we appreciate this positive feedback.

Fig. 2 & 3: I am not sure why these two figures are not combined into 1.

Although we agree that the original versions of these figures could have been combined into one, we felt that with the addition of new data to Figure 3 it might be better to keep the two separate.

Fig. 2: simply show neutrophils being able to form filopodia extension (which is already known). The only new data is Fig. 2A which show CA200645 at the base of the filopodia. One picture without any quantitation is not very telling. The authors should quantify how often they observed CA200645 at the base of filopodia.

We agree that quantification of plaque/cytoneme association was a crucial missing piece of information in the original version of this figure. We now have included this information in the figure legend and the text of the results section.

Fig. 4: p7. The authors postulated that "nanotubes" could "represent a mechanism for the targeted, long-range capture". They must add something regarding the similarities or differences with previous data on nanometer-thin micropodial extensions (NMEs) (Romero et al. *Cell Host & Microbe*, 2011).

Fig. 5.B: see general comment on forces.

We have incorporated comments relating to Romero et al.'s previous data into the manuscript; we believe that there are some very interesting parallels and that the added context will be very helpful for readers.

Overall, the authors do show that A₃ AR is important for the formation and function of these filopodia-like structures leading to the capture of bacteria. However, as is, I do not believe that the authors showed enough to warrant publication in this journal.

Based on the suggestions of you and the other reviewers, we have added data, clarified the novelty of our findings, and more carefully addressed how our findings fit into the "big picture" (particularly in the context of the work by Romero et al. and Galkina et al.). We hope that with these changes, you find our manuscript suitable for publication as a report-style article.

Referee #3

In this manuscript by Corriden et al, the authors use both qualitative and quantitative microscopic measures to evaluate the role of the neutrophil A₃-Adenosine Receptor in promoting neutrophil nanotube formation. The authors then expand their study by linking A₃AR-dependent nanotube formation to the ability of neutrophils to sense, tether, and reel in bacterial pathogens. In recent years, the A₃AR receptor has been shown to be important for neutrophil chemotaxis as influenced by ATP release. Additionally, the ability of neutrophils to elaborate long tubular extensions (spanning up to 100µm in length) capable of contacting/sensing bacterial pathogens has also been previously described for Salmonella enterica serovar Typhimurium. This manuscript elaborates on these prior studies by establishing a direct association between A₃AR cellular localization as well as A₃AR signaling and the enhanced production of neutrophil nanotubes capable of sensing bacteria. A₃AR mediated enhancement of nanotube formation implies a novel mechanism by which neutrophils can sense and respond to bacterial insult. In this reviewer's opinion, the work described herein is both novel and appealing. While the majority of the experimental design is sound, there are a number of points that should be addressed prior to publication.

1. Can the authors cite or provide evidence for both the potency and selectivity of the agonist 2-CI-IB-MECA? Because the authors draw important conclusions from the use of this agonist it seems essential to at least describe in some detail its degree of selectivity in the text.

We agree that some comment on the selectivity of 2-CI-IB-MECA is critical for providing context for readers. 2-CI-IB-MECA is extremely selective for the human A₃ receptor. It has a reported K_i value of 0.33 nM at rat A₃ receptors and is 2500- and 1400-fold selective for rat A₃ vs A₁ and A_{2A} receptors, respectively (Jacobson, 1998). We have also shown it to potently bind to the human adenosine A₃-receptor (Stoddart et al., 2012). A comment on the selectivity of this agonist and the A₃-selective antagonist MRS 1334 has now been added to the text.

2. On page 7, the authors state "a number of cells rapidly extended nanotubes in response to bacterial exposure". Can the authors provide a quantitative assessment of the percentage of neutrophils that elaborated nanotubes in response to bacterial exposure (i.e. are only a small subset of total neutrophils responding in this manner)? This is particularly important because in figures 3 and 4 the authors present quantitative data as nanotubes/100 cells. Unfortunately, quantifying in this manner does not give a precise indication of how many neutrophils are actually extending nanotubes (as the authors state, a single neutrophil is capable of elaborating multiple nanotubes). The data as presented in figures 3 and 4 would be acceptable only if the authors first provide a percentage of neutrophils extending nanotubes (independent of how many nanotubes are actually present on a single cell) compared to cells not extending nanotubes.

We agree that a quantitative assessment of the number of cells expressing cytonemes and the number of cytonemes per cell is important. We have added this information in the revised version of our manuscript.

3. In figure 3B, this reviewer wonders what percentage of nanotubes actually co-localize with A3AR both before and after 2-Cl-IB-MECA treatment. For example are the 20-30 nanotubes/100cells seen before agonist treatment independent of A3AR co-localization? If these "pretreatment nanotubes (no agonist)" are indeed associated with A3AR co-localization it is more difficult to directly associate nanotube formation with an A3AR function, in particular because MRS1334 treatment does not reduce nanotube formation below "pretreatment" levels. See #4 for a potential means by which the authors could experimentally address this point.

This is an important question, though one that is difficult to assess experimentally. Based on our observations, A3 antagonists such as MRS1334 do not cause these structures to retract; thus, and pre-existing cytonemes will persist even in the presence of such antagonists. Furthermore, it is difficult to avoid the spontaneous release of ATP and generation of extracellular adenosine from cells that are stimulated and/or damaged during the neutrophil isolation procedure. In the revised manuscript, we had made an effort to include a more quantitative assessment of the association of these structures and A3-AR plaques (see additions to the text and the Figure 2 legend). Even this quantification may be an underestimation of association; this is because we must use quite low concentrations of our fluorescent ligand (which is an antagonist), which may lead to incomplete staining.

4. In a previous study of R. Corriden (Ref. 3 in this manuscript), neutrophils from an A3AR KO mouse are used to evaluate the contribution of the A3AR receptor to neutrophil chemotaxis in response to ATP release. Are these mice accessible? If so, this reviewer feels it would be extremely valuable to assess nanotube formation using WT and KO murine neutrophils. This would help to shed light on to a number of ambiguities that exist in the manuscript regarding how often and in what capacity A3AR is associated with neutrophil nanotubes.

Although Ross Corriden was involved in a study that used the A3AR knockout mice, the mouse colony utilized for that study was maintained by a collaborator and no longer exists. Unfortunately, although we made a substantial effort to obtain these mice from Merck, we were unable to do so by the time our resubmission was due (please see our general comments above). In light of this, we have attempted to more carefully and quantitatively assess the association of A3ARs with nanotube-like structures, and have added additional functional data that more explicitly demonstrates a role of these receptors as facilitators of phagocytosis.

2nd Editorial Decision

24 May 2013

Thank you for your patience while your revised manuscript has been under peer-review. As you will see from the reports below, referees 1 and 3 are now positive about its publication in EMBO reports. Although referee 2 still has concerns, I think on the whole the study is now suitable for publication in here. I am therefore writing with an 'accept in principle' decision, which means that I will be happy to accept your manuscript for publication once the minor issues/corrections raised by referee 1 have been addressed.

In addition, in going through your study prior to publication, I have noted that information on the number of independent experiments and the identity of the error bars are missing from the legend to figure 5C, and the color coding in figure 4C seems to be switched.

Finally, as a standard procedure, we edit the title and abstract of manuscripts to make them more accessible to a general readership. Please find the edited versions below the referee reports and let me know if you do NOT agree with any of the changes.

Once all remaining corrections have been attended to, you will then receive an official decision letter from the journal accepting your manuscript for publication in the next available issue of EMBO reports. This letter will also include details of the further steps you need to take for the prompt inclusion of your manuscript in our next available issue.

Thank you for your contribution to EMBO reports.

REFEREE REPORTS:

Referee #1:

Although I would have like the authors to complete the requested work using A3AR null mice, I can appreciate their inability to gain access to those animals in a timely manner as they are currently housed at a pharmaceutical company. I think that they have done an adequate job addressing my other concerns.

Minor comment.

Pg 7 Fig 4D should be Fig 3D.

Pg 10. I think that "Connexin 46" should be "Connexin 26".

Referee #2:

Unfortunately I do not think that the authors have adequately replied to the referees concerns. The novelty of the manuscript is limited to the role A3AR , which has not been fully explored and mechanistically explained. I also do not see the new fig 4D.

In particular regarding my original comments, I do not find their quantization adequate, neither the reference to previous work satisfactory. It is not clear what is the difference between the structure they describe in neutrophils compared to the ones previously described in macrophage and epithelial cells.

In this context the relevance of their data appears quite limited and in my opinion do not warrant publication in EMBO Reports

Referee #3:

In this revised version the authors have adequately addressed the most pressing concerns raised about their study. Experiments performed with PMNs from the A3AR knock-out animals would have significantly substantiated the claims put forth here. However, the authors state that they no longer have access to these animals. Even with the lack of data from KO animals, in this reviewer's opinion, the work presented in the manuscript is significant, of broad interest and very exciting.

Edited title and abstract

Neutrophil A3-adenosine receptor microdomains promote the formation of bacteria-tethering cytonemes

The A3-adenosine receptor (A3AR) has recently emerged as a key regulator of neutrophil behavior. Using a fluorescent A3AR ligand, we show that A3ARs aggregate in highly-polarized immunomodulatory microdomains on human neutrophil membranes. In addition to regulating chemotaxis, A3ARs promote the formation of filipodia-like projections that can extend up to 100 μ m to tether and "reel in" pathogens. Exposure to bacteria or an A3AR agonist stimulates the formation of these projections and bacterial phagocytosis, whereas an A3AR-selective antagonist inhibits cytoneme formation. Our results shed new light on the behavior of neutrophils and identify the A3AR as a potential target for modulating their function.

2nd Revision - authors' response

04 June 2013

Recently, our manuscript, titled "Neutrophil A3-adenosine receptor microdomains promote the formation of bacteria-tethering cytonemes" was accepted in principle for publication in EMBO Reports by Nonia Pariente. As the first author of this publication, I have performed the manuscript preparation and submission, though I was out of the office for a short period of time when we received the notification of acceptance. I have now finished preparation of all of the final files for submission.

3rd Editorial Decision

06 June 2013

I am very pleased to accept your manuscript for publication in the next available issue of EMBO reports. Thank you for your contribution to our journal.

As part of the EMBO publication's Transparent Editorial Process, EMBO reports publishes online a Review Process File to accompany accepted manuscripts. As you are aware, this File will be published in conjunction with your paper and will include the referee reports, your point-by-point response and all pertinent correspondence relating to the manuscript.

If you do NOT want this File to be published, please inform the editorial office within 2 days, if you have not done so already, otherwise the File will be published by default [contact: emboreports@embo.org]. If you do opt out, the Review Process File link will point to the following statement: "No Review Process File is available with this article, as the authors have chosen not to make the review process public in this case."

Thank you again for your contribution to EMBO reports and congratulations on a successful publication. Please consider us again in the future for your most exciting work.